# Comparative Study on the Influence of Various Drying Techniques on Drying Characteristics and Physicochemical Quality of Garlic Slices

**DOI:** 10.3390/foods12061314

**Published:** 2023-03-20

**Authors:** Zhi-An Zheng, Shan-Yu Wang, Hui Wang, Hongmei Xiao, Zi-Liang Liu, Ya-Hui Pan, Lei Gao

**Affiliations:** 1College of Engineering, China Agricultural University, Beijing 100083, China; 2Henan Engineering and Technology Research Center for Garlic Deep Processing, Kaifeng 475200, China; 3College of Food Science and Technology, Nanjing Agricultural University, Nanjing 210095, China

**Keywords:** garlic quality, drying kinetics, microstructure, allicin, physicochemical characterization

## Abstract

Effects of vacuum freeze drying (VFD), air impingement drying (AID), hot air drying based on temperature and humidity control (TH-HAD), pulsed vacuum drying (PVD), and medium- and short-wave infrared radiation drying (MSIRD) on the drying characteristics and physicochemical properties of garlic slices were investigated in the current work. Based on the experimental results, the Weibull model fitted the experimental results better (*R*^2^ > 0.99) than the Wang and Singh model. Samples dried with PVD showed the smallest color difference (Δ*E**), better rehydration capacity and desirable reducing sugar content. In response to thermal effects and pressure pulsations, the cell walls gradually degraded, and the cell and organelle membranes ruptured. The allicin and soluble pectin contents of garlic slices treated with PVD were higher by 8.0–252.3% and 49.5–92.2%, respectively, compared to those of the samples dried by other techniques. VFD maintained a complete garlic slice structure with the minimum shrinkage and the best appearance. The MSIRD process produced the densest structure, and caused an additional loss of color and phytochemical contents. The findings in current work implied that PVD could be a promising drying technique for garlic slices.

## 1. Introduction

Garlic (*Allium sativum* L.), native to West and Central Asia, is widely used worldwide as a traditional medicinal and food homologous plant [1]. Data from the Food and Agriculture Organization indicate that global garlic production in 2021 was about 28.20 million tons, including China’s production of 20.51 million tons [2]. Besides being used as a condiment, garlic has many functions, such as treating cardiovascular diseases, regulating blood pressure, and improving the immune system [3,4], as well as showing anti-cancer, antimicrobial, and anti-tumor properties [5,6]. Containing allicin, sulfide, flavonoids, phenols, selenium, germanium, and other trace elements with antioxidant properties, garlic is used as a natural antioxidant [7].

Garlic has a short storage period. Generally speaking, it can be stored at 16 °C for 2 months. In addition, it can be preserved for 9 months at −1.5 °C and relative humidity of 60–70%. However, the cost and environmental impact of storage at low temperatures deserve attention [8,9]. Due to high moisture content, germination and decay seriously affect the nutritional, pharmacological and economic value of garlic if not protected in time [10]. Drying of high-moisture-content agricultural and food products could be effectively used to increase their shelf life; at certain moisture levels, the growth and multiplication of microorganisms as well as a great deal of moisture-mediated deterioration reactions are limited [11,12]. In recent years, dehydrated garlic has become more popular because of its storability and variety of good market prospects in products such as garlic slices, garlic granules, and garlic powder [13].

Currently, hot air drying (HAD) is continually used because of simple operation and low capital investment. Nevertheless, previous studies generally confirmed that HAD technology has the disadvantages of long drying time, which leads to quality deterioration of dried products such as color and flavor as well as the loss of nutrients [13,14]. Vacuum freeze drying (VFD) can relatively keep the color appearance and nutrient content of food well, but it is not applicable to industrial large-scale production due to its high energy consumption [15].

Air impingement drying (AID) technology uses a nozzle to generate high-speed hot air to directly impact the surface of materials for energy exchange [16]. Its convective heat transfer coefficient is considerably higher than that of HAD, with high efficiency and low energy consumption [17]. The technology has been effectively applied to mushroom slices [18], potato cubes [19], red cabbage [20], etc. Hot air drying based on temperature and humidity control (TH-HAD) is a new technology for drying agricultural materials by regulating the values of temperature (T, °C) and relative humidity (RH, %) of the drying medium during the moisture removal process [21]. It can reduce the shrinkage of material by increasing the RH of the medium, prevent the surface of the material from deteriorating and alleviate the oxidation reaction of heat-sensitive materials during the drying process [22]. In view of these advantages, TH-HAD has been primarily employed in the drying of long-grain rice [23], longan [24], shiitake mushroom [25], etc. Pulsed vacuum drying (PVD), is a relatively novel dehydration technology that uses sequential change in vacuum pressure in the dryer interior to enhance moisture translocation and quality of materials during the dehydration process [26]. The samples are kept in a vacuum condition for quite a while during the process, which can reduce the occurrence of oxidative browning of the nutrients in the material [27]. PVD has been successfully applied to materials containing heat-sensitive components, such as mango [27], *Ginkgo biloba* L. seed [28], ginger [29], etc. Medium- and short-wave infrared radiation drying (MSIRD) is a novel dehydration technology with high efficiency. Compared with convection drying, infrared radiation drying could cut energy consumption by 50% [30]. MSIRD has been commendably practiced to dry different products such as *Stevia* leaves [31], soymilk residues [32], sponge gourd slices [33], and tangerine peels [34].

Considering the nutritive and medicinal value of garlic and the necessity of proper processing, appropriate drying technology can reduce dehydration time and improve the quality of dried garlic slices. Therefore, in the current study, the influence of various drying methods (VFD, AID, TH-HAD, PVD, and MSIRD) on the drying characteristics and physicochemical characteristics (rehydration ratio, color, allicin, soluble pectin and reducing sugar content) and microstructure of garlic slices were studied and discussed. The outcomes of this research work will be conducive to finding an optimized drying method for garlic slices and provide a theoretical basis for improving the production efficiency and quality of garlic slices.

## 2. Materials and Methods

### 2.1. Raw Materials

The same batch of unpeeled fresh garlic was supplied by Qixian Panan Food Co., Ltd. (Kaifeng, China), and stored in a refrigerator (4 ± 1 °C, RH of 90%) for no more than 1 week. In order to ensure the uniformity of the material, garlic cloves with no mechanical damage and uniform size were selected randomly after peeling. After rinsing with tap water, the surface moisture of the garlic cloves was absorbed with soft paper, and the cloves were sliced to 4.0 ± 0.1 mm thickness perpendicular to the growth direction. The initial moisture content of the garlic slices was measured by vacuum drying at 70 °C for 24 h, following the AOAC standard method [35], and determined to be 62.15% ± 1.63% (wet basis, w.b.).

### 2.2. Drying Experiments

In this study, to evaluate the effect of five different drying methods on the drying characteristics and quality of garlic slices, based on the results of previous experiments, all of the dehydration experiments were carried out at constant temperature of 60 °C. Weight loss of garlic slices was determined with an electronic balance (SP402, Ohaus Co., Ltd., Parsippany, NJ, USA) with an accuracy of ±0.01 g, and the garlic slices were dehydrated until the terminal moisture content of 6% (w.b.). All of the drying experiments were repeated three times.

#### 2.2.1. Vacuum Freeze Drying (VFD)

Approximately 120 g of garlic slices was pre-frozen in a vacuum freeze drying apparatus (LGJ-25C, Sihuan Scientific Instrument Co., Ltd., Beijing, China) at −50 °C for 3 h, then placed in the same dryer at an absolute pressure, shelf temperature and condensing temperature of 20 Pa, 30 °C and −50 °C, respectively.

#### 2.2.2. Air Impingement Drying (AID)

The air impingement drying equipment (located in the College of Engineering of China Agricultural University, Beijing, China; and previously described by Liu et al. [18]) was used to perform AID experiments (Figure 1). According to preliminary experiments, the air velocity of the dryer was selected to be 6 m/s. About 150 g of the garlic samples was spread out evenly on the stainless-steel plate.

#### 2.2.3. Hot Air Drying Based on Temperature and Humidity Control (TH-HAD)

Figure 2 shows a schematic view of the used hot air dryer with integrated temperature and relative humidity control (installed in the College of Engineering of China Agricultural University, Beijing, China; described by Wang et al. [19] previously). Based on the results of previous experiments, the relative humidity of the equipment was controlled at 16–20%, and about 150 g of garlic slices was spread evenly on a stainless-steel material tray.

#### 2.2.4. Pulsed Vacuum Drying (PVD)

To conduct the PVD experiments, the pulsed vacuum dryer installed in the College of Engineering of China Agricultural University, Beijing, China was used. A structure diagram of the PVD equipment is shown in Figure 3, and detailed properties have been provided by Wang et al. [36]. The drying system is mainly composed of three units, consisting of a vacuum system, a heating system and an electronic control system. The electronic control system could automatically adjust the drying conditions according to the parameters set by the user. Based on previous experiments, the vacuum holding time of the equipment was 13 min, and the atmospheric holding time was 3 min. Approximately 600 g of sample was evenly spread on the stainless-steel tray for each drying experiment.

#### 2.2.5. Medium- and Short-Wave Infrared Radiation Drying (MSIRD)

A medium- and short-wave IR-convective dryer (Senttech Infrared Science and Technology Co., Ltd., Taizhou, China; Figure 4) previously described by Zhang et al. [33] was used to conduct the MSIRD experiments. Three infrared lamps with different radiant power including a medium-wave (2–4 μm) infrared lamps (450 W) and two short-wave (0.75–2 μm) infrared lamps (450, 450 W) were arranged parallel to each other in the dryer. Moreover, six rows of 18 annular airflow nozzles were installed above the infrared lamps to achieve strong convection of radiant heat with air velocity of 3 m/s. The experimental radiation distance between the infrared lamps and the material surface was adjusted to be about 13 cm. For each drying experiment, approximately 150 g of the garlic slices was spread evenly on the stainless-steel tray.

### 2.3. Drying Characteristics

#### 2.3.1. Drying Kinetics

The drying curve of garlic samples during the dehydration process was based on the variation of moisture ratio (*MR*) with drying time. The *MR* of garlic slices at various drying times was expressed as shown in the following equation [37]:(1)$$MR=\frac{M_{t}-M_{e}}{M_{0}-M_{e}}$$
where *M*_0_ and *M_t_* are the initial and dry basis moisture content of the material at drying time 0 and *t*, respectively, g/g (d.b.); *M_e_* is the equilibrium moisture of the dried product, g/g (d.b.).

As the *Me* value of garlic slices was much less than *M*_0_ and *M_t_*, Equation (1) could be simplified as follows:(2)$$MR=\frac{M_{t}}{M_{0}}$$

The drying rate (*DR*) was calculated by the following equation [28]:(3)$$DR=\frac{M_{t1}-M_{t2}}{t_{2}-t_{1}}$$
where *t*_1_ and *t*_2_ are the drying times, min; *M_t_*_1_ and *M_t_*_2_ are the dry basis moisture content of the material at drying times *t*_1_ and *t*_2_, respectively, g/g (d.b.).

#### 2.3.2. Modeling of Drying Curves

The Weibull model has been widely used in recent years in the fields of agricultural processing, pharmacology, and mechanical engineering due to its simplicity, flexibility, and great applicability [24]. The moisture change and drying characteristics during drying can be indicated by combining the scale parameter *α* and shape parameter *β* of the Weibull model with the material [38]. The Weibull equation depicted by Cunha et al. [39] was as follows:(4)$$MR=\exp\left[-{\left(\frac{t}{\alpha }\right)}^{\beta }\right]$$
where *α* is the scale parameter, indicating the rate constant during the drying process, min; *β* is the shape parameter.

In addition, for comparison with the Weibull model, the Wang and Singh model, which is an empirical model commonly used to describe moisture changes during the drying of agricultural products, was chosen and calculated by Equation (4) according to Wang and Singh [40]:(5)$$MR=1+at+bt^{2}$$
where *a* and *b* are the drying constants; *t* is the drying time, min.

The performance of the two models was evaluated using a wide range of statistical parameters such as the chi-square (*χ*^2^) parameter, residual sum squares (*RSS*), and coefficient of determination (*R*^2^), which were determined by the following equations [41,42]:(6)$$\chi^{2}=\frac{\sum_{i=1}^{N}{\left(MR_{\mathrm{pre},i}-MR_{\exp,i}\right)}^{2}}{N-n}$$
(7)$$RSS=\sum_{i=1}^{N}{\left(MR_{\mathrm{pre},i}-MR_{\exp,i}\right)}^{2}$$
(8)$$R^{2}=1-\frac{\sum_{i=1}^{N}{\left(MR_{\mathrm{pre},i}-MR_{\exp,i}\right)}^{2}}{\sum_{i=1}^{N}{\left(\bar{MR_{\mathrm{pre},i}}-MR_{\exp,i}\right)}^{2}}$$
where *MR*_pre,i_ indicates experimental value and *MR*_exp,i_ represents predicted value; *N* means the amount of experience, and *n* is the number of constants.

### 2.4. Rehydration Ratio (RR)

Rehydration experiments for the dried garlic slices were carried out in accordance with the procedure described by Zhou et al. [43] with slight modifications. The dried sample (10 g) was added to 200 mL of distilled water and kept at 35 °C for 1 h, then filtered through filter paper to absorb the surface water and weighed using an electronic balance. The following equation was used for calculating the rehydration ratio [44]:(9)$$\mathrm{RR}=\frac{W_{w}}{W_{d}}$$
where *W_w_* and *W_d_* are the mass (g) of the garlic slices after and before rehydration, respectively.

### 2.5. Microstructure Measurement

The dried samples were cut (2 mm × 2 mm) using a scalpel and fixed on a carrier plate with double-sided tape. Then, the plate was placed in a vacuum ion sputter coater for gold spraying for 50 s. A scanning electron microscope (SU3500, Hitachi, Tokyo, Japan) was used to observe the surface microstructure of garlic slices under an accelerating voltage of 15 kV.

### 2.6. Color

The reflectance mode of a LabScan XE spectrophotometer was used to measure the *L**, *a**, and *b** values of the different dried samples. *L** represents the bright value and varies from 0 (black) to 100 (white); *a** represents the green–red value and ranges from −60 (pure green) to 60 (pure red); and *b** represents the blue–yellow value and ranges from −60 (pure blue) to 60 (pure yellow). The total color difference (Δ*E**) between the dried and the fresh garlic slices was calculated as follows [45]:(10)$$\Delta E^{*}={\left[{\left(L^{*}-L_{0}^{*}\right)}^{2}+{\left(a^{*}-a_{0}^{*}\right)}^{2}+{\left(b^{*}-b_{0}^{*}\right)}^{2}\right]}^{1/2}$$

### 2.7. Allicin Content

The content of allicin was determined using the method described by Qiao et al. [44] with slight modifications. Garlic powder (sieved through 80 mesh, 0.3 g) and distilled water (8 mL) were mixed thoroughly in a test tube, allowed to stand for 15 min, and then centrifuged (4000 r/min, 15 min). The supernatant (1 mL) was incubated with cysteine solution (10 mmol/L, 5 mL) at 26 °C for 15 min. The reaction mixture (1 mL) was placed into a volumetric flask (100 mL) and fixed with distilled water. Finally, the reaction mixture (4.5 mL) was diluted 100 times with DTNB solution (5,5-Dithiobis, 1.5 mmol/L, 0.5 mL), incubated at 26 °C for 15 min, and the absorbance was measured at 412 nm. The content of allicin was calculated according to the following equation [44]:(11)$$c=\frac{\left|A_{0}-A\right|\times \beta \times 162.27}{2\times 14,150\times 100}$$
where *c* expresses the concentration of allicin (mg/g), *A*_0_ means the absorbance of cysteine solution after reaction with DTNB solution, *A* represents absorbance of the reaction mixture of garlic extract and cysteine solution after reaction with DTNB solution, *β* is the dilution factor, 162.67 indicates the molar mass of allicin (g/mol), and 14,150 is the molar extinction coefficient of 2-nitro-5-thiobenzoic acid (NTB) at 412 nm.

### 2.8. Soluble Pectin Content

The soluble pectin content was measured in accordance with the description by Cao et al. [46] with a few modifications. A volume of 0.5 mL of the extract was aspirated; distilled water (0.5 mL) and concentrated sulfuric acid (6 mL) were added, and the mixture was heated in a boiling water bath for 20 min, then removed and cooled at room temperature. Then, carbazole-ethanol solution (1.5 g/L, 0.2 mL) was added and the mixture was shaken well. After being left in the dark for half an hour, the absorbance value of the reaction solution at 530 nm was measured using a UV spectrophotometer (Pursee General Instrument Co., Beijing, China).

### 2.9. Reducing Sugar Content

The garlic polysaccharide content was determined in accordance with the method of Cao et al. [46] with a few modifications. The extract (2 mL) and 3,5-dinitrosalicylic acid reagent (1.5 mL) were mixed well. The mixture was heated for 5 min in a boiling water bath, removed and immediately cooled to room temperature with cold water, and then fixed in distilled water to 25 mL and mixed well. The absorbance value of the reaction solution was measured at 540 nm by a UV spectrophotometer.

### 2.10. Statistical Data Analysis

Experimental data were statistically compared and analyzed by IBM SPSS Statistics (version 27.0, IBM Corp., Armonk, NY, USA), Origin 2022 (OriginLab Corp., Northampton, MA, USA), and Microsoft Excel 2016 (Microsoft Corporation, Redmond, WA, USA). All measurements were repeated at least three times, and one-way analysis of variance (ANOVA) was performed using Duncan multiple test (*p* < 0.05). The final results were presented as mean ± standard deviation (SD).

## 3. Results and Discussion

### 3.1. Drying Characteristics Analysis

#### 3.1.1. Drying Kinetics Curves

Moisture removal kinetics of the garlic slices under the various practiced drying techniques are displayed in Figure 5. From Figure 5a, the different drying techniques had a significant (*p* < 0.05) effect on the drying times at the same drying temperature (60 °C). The drying time of garlic slices was shortest for AID (taking only 183 min), followed by TH-HAD (225 min) and MSIRD (233 min), while PVD had the longest drying time (316 min). This can be attributed to the fact that AID technology has the advantages of high air velocity, short flow velocity, and a thin boundary layer between the sample surface and air, which leads to a high convective heat transfer coefficient, thus achieving high drying rates [47]. In the case of AID drying, the higher heat transfer coefficient results in more heat gain in a shorter period and accelerates the moisture evaporation [48,49].

Drying rate curves for various thermal drying methods are shown in Figure 5b. Obviously, at the same temperature (60 °C), the drying rates of the AID and TH-HAD techniques were significantly (*p* < 0.05) higher than those of MSIRD and PVD. Moreover, during AID and TH-HAD, the drying rate was decreased with the moisture content reduction. The entire dehydration of the garlic slices occurred at decreasing rate period, revealing that the internal moisture diffusion is the main phenomenon controlling the drying process. During the TH-HAD drying process, the internal moisture of the sample was quickly removed in the initial stage, under the condition of high humidity in the drying chamber [21]. Through the MSIRD process, the absorption of infrared radiation would quickly generate heat on the surface and inner layers of the samples, which may have facilitated the moisture removal [33]. At the same time, the drying rate of the garlic slice surface was higher than that of the inner layers, leading to the rapid shrinkage of the samples in the early stage of drying and affecting the internal moisture migration. For PVD, the drying rate was the lowest. This may be due to the pressure relief valve that remained open during the atmospheric operation of the PVD, allowing cold air into the drying chamber as well as pumping a large amount of heat around during the vacuum holding stage by the vacuum pump, both of which decreased the drying chamber and sample temperature and consequently reduced the drying rate of the samples [27]. Similar results have been also expounded by Deng et al. [50] and Meng et al. [51], who applied PVD to dry red pepper and yam slices, respectively.

#### 3.1.2. Modeling of Drying Curves

As presented in Table 1, the thermal drying data for garlic slices were fitted using the Weibull model and Wang and Singh model. The fit of the Weibull model was better than that of the Wang and Singh model, as can be seen by comparing the *R*^2^ of the two models. For the Weibull model, *α* was the scale parameter indicating the time required to evaporate 63% of the moisture in the material, ranging from 30.4653 to 100.7441 min. The lower its value, the shorter the drying time will be [52]. The range of values for the shape parameter *β* was from 0.92 to 1.39. When *β* was between 0.3 and 1 (i.e., AID and TH-HAD), it illustrated that the drying process was controlled by internal moisture diffusion at a falling rate. When *β* > 1 (i.e., PVD and MSIRD), the closer the value of *β* to 1, the higher the drying rate at the beginning [24]. This result is in accordance with the drying rate curves as depicted in Figure 5b.

### 3.2. Rehydration Ratio and Microstructure

Rehydration capacity refers to the degree to which dried products are restored to their original fresh state after rehydration, and it is an essential indicator reflecting the structural changes of dried products [53]. The rehydration ratios of dried garlic slices under different drying techniques are shown in Figure 6. Interestingly, the samples dried using the VFD technique had the lowest rehydration capacity. Similar results were reported by Feng et al. [54], who found significant amounts of intercellular air in garlic samples after VFD. This phenomenon may be explained by the fact that the rehydration process for the dried garlic slices is an exchange process between the intercellular air of the sample and the external water. When huge quantities of intercellular air in the VFD samples were exchanged with water, the water potential was not sufficient to deplete all of the air within the tissues. This phenomenon resulted in air occupying extensive intercellular space and impeded the uptake of water into the samples. This finally reflected that the VFD samples had a low rehydration ratio. There was a spot of intercellular air in the rehydrated thermal dried samples that might be due to the compact structure and less intercellular space formed by volume shrinkage during dehydration [54]. In addition, when comparing garlic samples with different thermal drying techniques, there was a significant difference (*p* < 0.05) in the rehydration capacity of the MSIRD samples compared to the other three dried samples. For MSIRD, the high infrared radiation intensity irradiated on the surface of the garlic slices might cause higher thermal damage to the sample, leading to irreversible changes in microstructure and a worse rehydration ratio [19].

Microstructure is an indispensable parameter to comprehend potential cellular mechanisms and observe the spatial morphology and distribution of cell walls more clearly to improve the drying systems performance [55]. As shown in Figure 7, the microstructure images of samples under the different drying techniques showed a porous structure. The VFD samples had the largest average pore diameter and retained a more intact cellular structure. Combined with the images of garlic slices in Table 2, it could be found that compared with the fresh slices, the dried slices had different degrees of shrinkage, among which VFD slices showed the smallest shrinkage. The AID, TH-HAD and PVD samples also maintained a compact and porous structure with better rehydration capacity. For MSIRD, the sample structure showed more severe collapse and deformation with the most severe cell wall damage and poor rehydration. Deng et al. [50] discovered a similar phenomenon. This might be due to the absorption of infrared energy on the surface of the garlic samples and the faster evaporation of moisture from the garlic tissue at high temperatures, resulting in cell deformation and collapse. The phenomenon of microstructure was consistent with the results for the rehydration ratio.

### 3.3. Color and Form

The values of *L**, *a**, *b**, and Δ*E* and corresponding images are tabulated in Table 2. The a* value increased significantly (*p* < 0.05) after drying. This may be due to the degradation or transformation of the pigments. VFD samples had the highest *L** and lowest *b** compared to the other samples. In fact, VFD effectively improved the brightness and whiteness of the garlic slices. It is possible that the continuous vacuum and the low thermal damage induced by the low-temperature environment slowed or inhibited the onset of browning. In general, the color difference can be clearly detected by the observer when Δ*E* is higher than 5. Lower Δ*E* values mean less color change during drying, indicating that the color of the PVD samples was closer to that of the fresh samples [56,57]. The MSIRD samples had the lowest *L**, the highest *a** and the highest Δ*E*. This is probably due to the high absorption of IR energy by the samples and non-uniform heating, resulting in a severe browning reaction.

For images of the garlic slices (Table 2), in contrast to VFD, the thermal drying technique caused severe shrinkage of the sample volume, probably due to the rapid evaporation of surface moisture, which created a pressure difference between the exterior and interior of the material. As a result, the entire matrix was dragged toward the center, leading to significant changes in the shape and dimensions of the material [54].

### 3.4. Allicin Content

Allicin is responsible for the pungent smell of garlic and is a major bioactive component in garlic products that is formed by a series of reactions of alliin (mainly distributed in cytoplasm) in the presence of alliinase (distributed in vesicle) after the garlic clove structure is crushed [58,59]. Statistical analysis substantiated that the allicin content of dried samples was significantly (*p* < 0.05) affected by different drying methods, as shown in Figure 8. The allicin was found to be better retained with PVD, TH-HAD, and AID, at 12.71, 11.77, and 10.44 mg/g, respectively. Poor retention of allicin occurred with VFD (5.37 mg/g) and MSIRD (3.61 mg/g). The pattern of variation demonstrated a significant (*p* < 0.05) correlation with the rehydration capacity and microstructure of garlic slices. This may be related to the fact that desiccation affects the permeability of plant cell and organelle membranes, resulting in the destruction of allicin and the partial inactivation of alliinase, interrupting further synthesis of allicin. Similar results have been reported for allicin retention during garlic drying by Aware and Thorat [60] and Ratti et al. [61].

### 3.5. Soluble Pectin (SP) and Reducing Sugar (RS) Content

A close relationship exists between the change in color quality of agro-products and the internal pectin content as well as reducing sugar content. The contents of soluble pectin (SP) and reducing sugar (RS) in the garlic slices dried under various methods are shown in Figure 9, ranging from 5.28% to 10.15% and 28.31 to 31.01 mg/g, respectively. For SP, the highest retention rate was achieved with the PVD technique, which may be conducive to the alternating vacuum and atmospheric pressure in the drying environment. In this case, the cellular structure of garlic was disrupted, and the chances of catalytic hydrolysis occurring in contact with cellulase, polygalacturonase and the corresponding substrate were increased. This phenomenon leads to a shortening of the cellulose and pectin molecular chains and results in an increase in SP [62]. For RS, the retention levels in the samples dried under VFD and PVD were significantly (*p* < 0.05) different from those of samples dried under the other three dehydration technologies. This is probably due to the gradual consumption of reducing sugars as substrates participated in the Maillard reaction during the drying process. In contrast, the VFD and PVD were in a full vacuum and vacuum alternating with the atmosphere environment, respectively, which greatly decreased the contact between RS components and oxygen and weakened the consumption of reducing sugars and inhibited browning [63]. This observation is closely related to color. In summary, the dried samples under PVD showed the best retention of SP and RS.

## 4. Conclusions

In view of the results revealed in this research, it is clear that drying technologies significantly (*p* < 0.05) affected the drying characteristics and physicochemical properties of garlic slices and agreed well with the Weibull model (*R*^2^ > 0.99). VFD technology was able to preserve the appearance properties of the garlic slices and the RS content. For thermal drying techniques, SEM results showed that AID, TH-HAD and PVD formed porous structures and had superior rehydration capacity. In addition to having the shortest drying process duration, AID yielded the same results as TH-HAD in terms of physicochemical properties. Meanwhile, the PVD-dried samples showed the lowest color difference compared to fresh samples and the best retention of allicin, SP and RS; however, the drying time required for PVD was the longest. The current results indicated that the PVD technique resulted in the best-quality products in comparison with the other studied drying methods. Future research should be performed to focus on optimizing the drying process parameters to reduce drying time and energy consumption.

## Figures and Tables

**Figure 1 foods-12-01314-f001:**
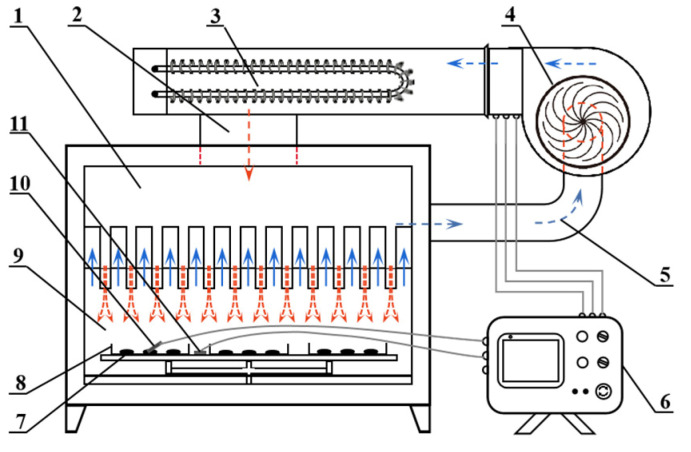
Structure diagram of air impingement drying equipment (The arrows indicate the direction of airflow). (1) Airflow distribution chamber, (2) drying air channel, (3) electric heater, (4) centrifugal fan, (5) drying air recycle channel, (6) temperature sensor of the controller, (7) sample, (8) sample tray, (9) drying chamber, (10) temperature sensor, (11) air velocity sensor.

**Figure 2 foods-12-01314-f002:**
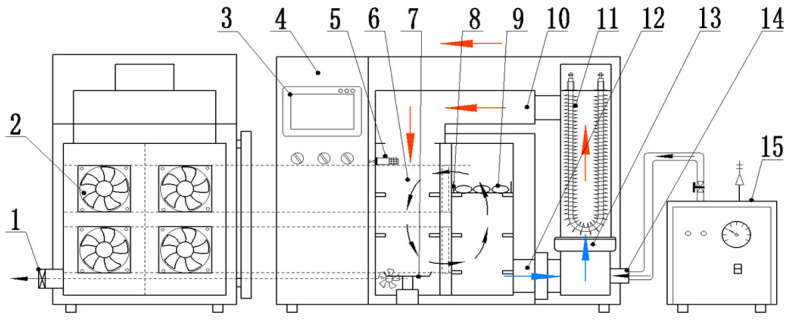
Schematic diagram of temperature and humidity controlled hot air dryer (The arrows indicate the direction of airflow). (1) Dehumidification axial fan, (2) disturbing fan, (3) touch screen, (4) control cabinet, (5) temperature and humidity sensor, (6) drying chamber, (7) weighing module, (8) material tray, (9) samples, (10) air bellows assembly, (11) electric heating tube, (12) return air duct, (13) axial flow fan, (14) steam inlet, (15) steam generator.

**Figure 3 foods-12-01314-f003:**
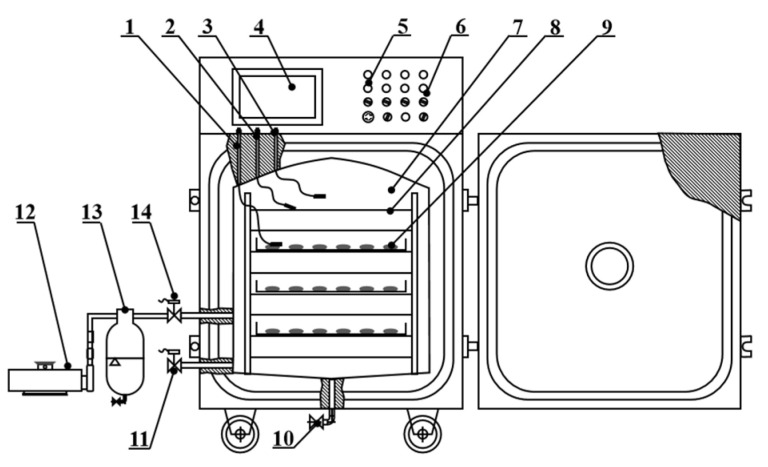
Structure diagram of pulsed vacuum drying equipment. (1) Material temperature sensor, (2) infrared-board temperature sensor, (3) pressure sensor, (4) touch screen control panel, (5) indicator light, (6) adjusting switch, (7) drying chamber, (8) far-infrared radiation heating element, (9) samples, (10) drain solenoid valve, (11) air solenoid valve, (12) vacuum pump, (13) condenser, (14) vacuum valve.

**Figure 4 foods-12-01314-f004:**
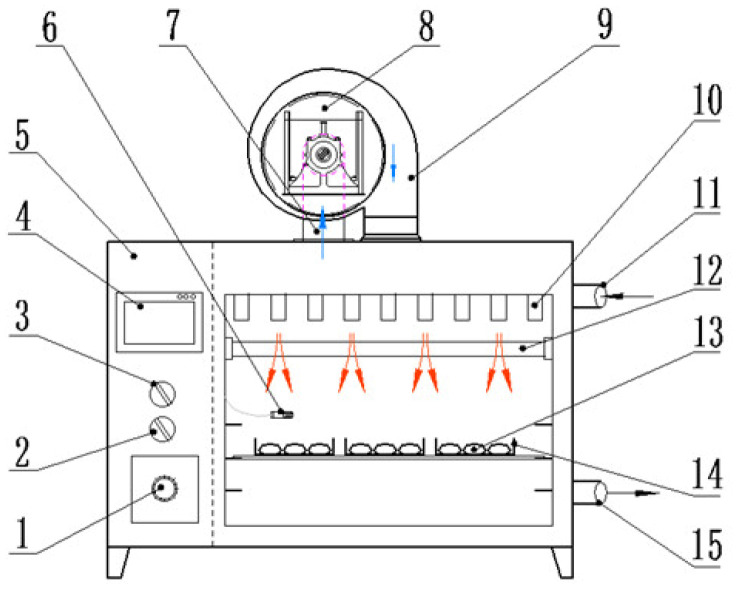
Structure diagram of medium- and short-wave infrared drying equipment (The arrows indicate the direction of airflow). (1) Fan switch, (2) power supply control switch, (3) lamps switch, (4) touch screen, (5) control cabinet, (6) temperature sensor, (7) return air duct, (8) centrifugal fan, (9) air inlet duct, (10) spray nozzle, (11) air inlet port, (12) infrared heating tubes, (13) samples, (14) stainless-steel tray, (15) wet discharging port.

**Figure 5 foods-12-01314-f005:**
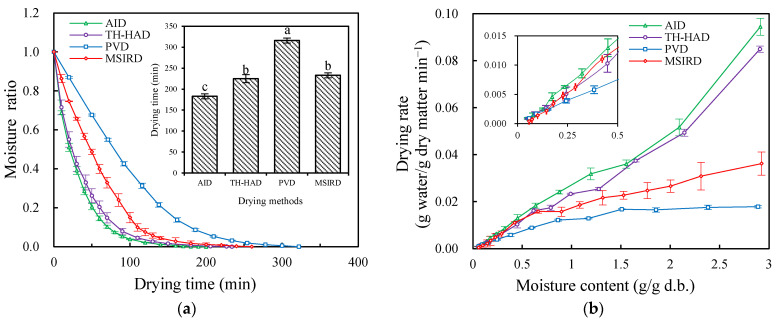
Moisture ratio curves and total drying time (**a**) and drying rate curves (**b**) of garlic slices under different thermal drying methods. The different letter (a–c) indicates statistically significant difference at *p* < 0.05.

**Figure 6 foods-12-01314-f006:**
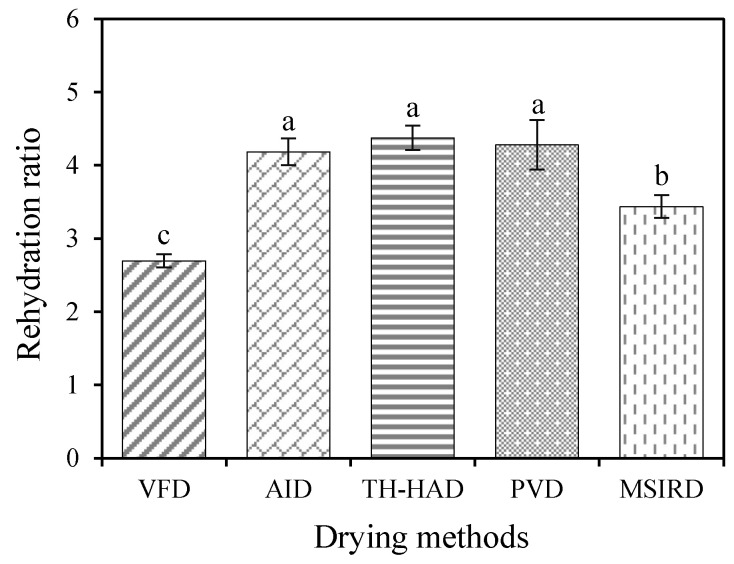
The rehydration ratios of dried garlic samples under various drying methods. The different letter (a–c) indicates statistically significant difference at *p* < 0.05.

**Figure 7 foods-12-01314-f007:**
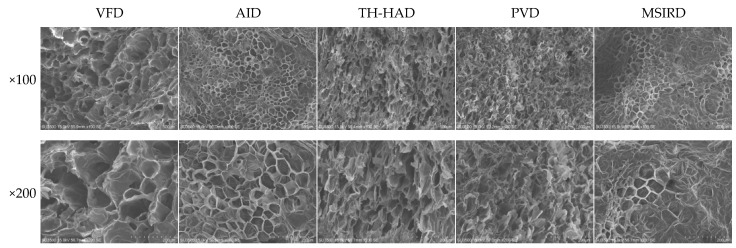
The microstructure of garlic slices dried in different methods at 100 and 200 magnifications.

**Figure 8 foods-12-01314-f008:**
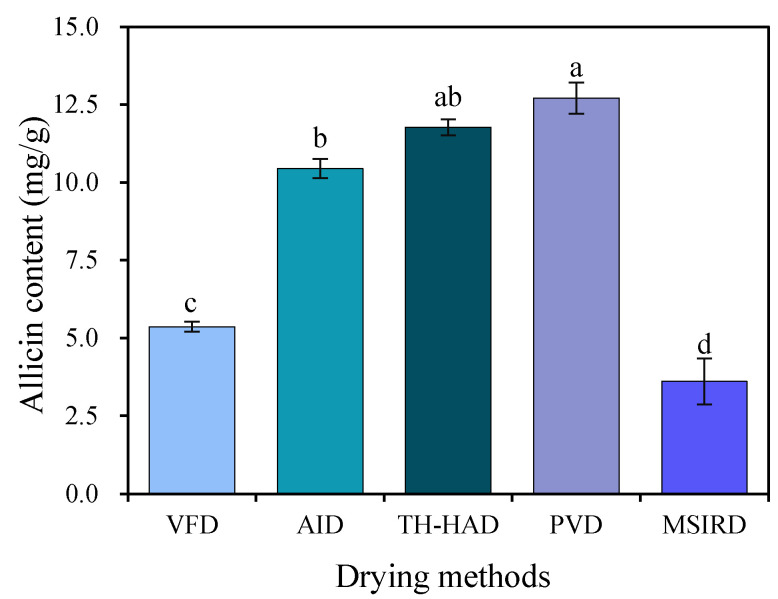
The allicin content of garlic slices dried in different methods. The different letter (a–d) indicates statistically significant difference at *p* < 0.05.

**Figure 9 foods-12-01314-f009:**
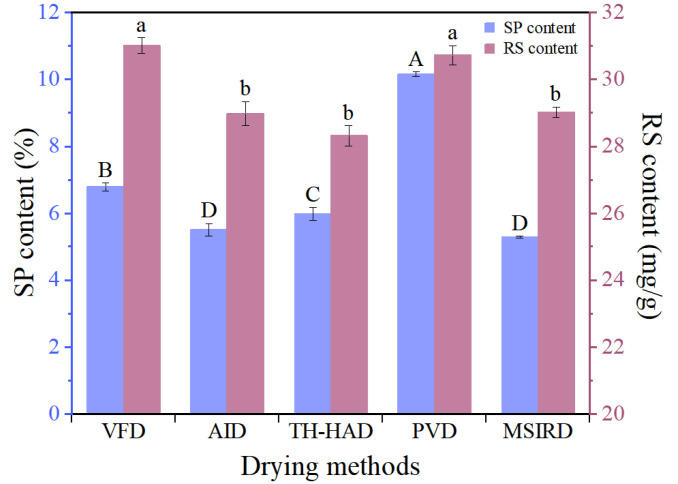
The soluble pectin (SP) and reducing sugar (RS) content in garlic slices dried using different methods. For SP and RS content, the different letters (A–D, a,b) indicate statistically significant difference at *p* < 0.05, respectively.

**Table 1 foods-12-01314-t001:** Modeling of drying curves under various drying methods.

Drying Methods	Weibull					Wang and Singh				
	*α* (min)	*β*	*χ* ^2^	*RSS*	*R* ^2^	*a*	*b*	*χ* ^2^	*RSS*	*R* ^2^
AID	30.4653	0.95	7.20 × 10^−5^	0.001	0.999	−0.0167	6.19 × 10^−5^	0.0128	0.204	0.850
TH-HAD	35.3494	0.92	1.00 × 10^−4^	0.001	0.999	−0.0145	4.58 × 10^−5^	0.0159	0.223	0.833
PVD	100.7441	1.39	1.54 × 10^−4^	0.002	0.999	−0.0074	1.37 × 10^−5^	0.0003	0.003	0.998
MSIRD	61.2684	1.25	3.80 × 10^−4^	0.007	0.997	−0.0112	2.95 × 10^−5^	0.0019	0.034	0.983

**Table 2 foods-12-01314-t002:** Images and color parameters of garlic slices under various drying methods.

	Fresh	VFD	AID	TH-HAD	PVD	MSIRD
Image	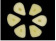	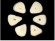	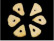	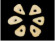	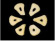	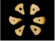
*L**	79.36 ± 0.68 ^d^	91.57 ± 0.41 ^a^	82.93 ± 0.17 ^c^	83.52 ± 0.28 ^c^	87.98 ± 0.03 ^b^	74.97 ± 0.51 ^e^
*a**	−5.86 ± 0.81 ^e^	−2.35 ± 0.51 ^d^	2.57 ± 0.32 ^b^	2.47 ± 0.06 ^b^	−1.03 ± 0.14 ^c^	4.67 ± 0.21 ^a^
*b**	7.98 ± 0.27 ^c^	3.34 ± 0.18 ^d^	24.22 ± 0.43 ^a^	23.93 ± 0.36 ^a^	13.28 ± 0.20 ^b^	24.18 ± 0.40 ^a^
Δ*E*	—	13.53 ± 0.44 ^c^	18.64 ± 0.50 ^b^	18.47 ± 0.27 ^b^	11.21 ± 0.17 ^d^	19.82 ± 0.31 ^a^

Note: In the same row, various letters (a–e) indicate significant differences (*p* < 0.05) in conformity with Duncan’s test.

## Data Availability

The data presented in this study are available on request from the corresponding author.

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
