# Peer review of "Comparative Study on the Influence of Various Drying Techniques on Drying Characteristics and Physicochemical Quality of Garlic Slices"

_foods, 2023, doi:10.3390/foods12061314_

Round 1

Reviewer 1 Report

The manuscript by Zheng et al compares the 5 novel drying technologies with respect to drying of garlic slices. The hypothesis is well explained. The manuscript has some interesting findings. The language is good and easy to understand.  

I have following observations to improve its quality as-

Keywords: have scope for improvement as replace color, soluble pectin with more suitable one

L18: attractive means desirable?

L30: please cite a recent data if available

L33: resisting bacteria means antimicrobial properties

L36: natural biological antioxidant, plz delete biological

L37 : short storage period; please mention the period

L48: use appropriate term for worse loss of nutrients or delete worse

L77: edible value may be nutritive value

L278: relevas, plz correct

L 296: please rewrite

Tables: footnote is required

Reviewer 2 Report

foods-2217592

Comparative study on the influence of various drying techniques on drying characteristics and physicochemical quality of garlic slices

Introduction:

Line 28 – “Native from West and Central”

Line 30 – maybe it is better to represent in million tones than in scientific notation.

Line 33 – What do you mean with “resisting bacteria”?

Material and methods

Line 91 – How do you define hard texture? And how did you measure this? Pressing?

Line 99 – “results of previous experiments”

Line 108-109 – What means/represent this heating temperature?

Line 171 – What was the equilibrium moisture used/how did you determine this parameter?

Line 229 – This expression of allicin in mg/g was made in relation to dry basis?

Line 236 and 244 – what are these extracts mentioned?

Results and Discussion

Line 271 – Figure 5 – The Figure need to be improved. The subtitles indicating each treatment is missed.

How is possible to scale up this technologies for industrial applications? Or what technology is more suitable for this?

IN relation to the energy consumption, how to compare these technologies?

Reviewer 3 Report

The work "Comparative study on the influence of various drying techniques on drying characteristics and physicochemical quality of garlic slices" has been reviewed with the following comments

I consider that the article is of high quality, by virtue of the approach of the research objective and the relevant results obtained. I particularly consider that the article is very well developed, except for some small details in the graphics.

The presentation of the results as well as their discussion were appropriate. However, comparing these results with some others obtained around garlic and explaining them from food chemistry what their scope is could improve this article, which is already very good.

Small observations were made on the document attached to this review.

I take this opportunity to congratulate the authors for such a good contribution.
